# OpenReview forum: "GraphFractalNet: A Fractal-Inspired Sparse Transformer for Ultra-Scalable Graph Representation Learning"
_ICLR.cc/2026/Conference — ICLR 2026 Conference Withdrawn Submission_

### Official Review · Reviewer_zgCF · 2025-10-26

**Soundness:** 1
**Presentation:** 2
**Contribution:** 2
**Rating:** 2
**Confidence:** 3

**Summary:**

The paper introduces \textit{GraphFractalNet}, a graph neural network that integrates spectral graph representations, hierarchical clustering, and dynamic edge rewiring through a proposed fractal attention mechanism.
Each layer operates in a Laplacian spectral space and selectively rewires edges using a distance-aware score based on shortest-path distances.
The authors claim a per-layer complexity of $\mathcal{O}(N \log\log N)$ and provide theoretical arguments that the model is strictly more expressive than the $k$-Weisfeiler-Lehman (k-WL) test for $k \ge 2$, supported by a generalization bound.
Empirical results on a wide range of standard benchmarks (ZINC, OGBN, etc.) show state-of-the-art performance and improved scalability compared to prior graph Transformer architectures.

While the empirical resutls are impressive, the theoretical claims in expressiveness and complexity seem deeply flawed and, in my reading, do not hold.

**Strengths:**

I like the conceptual ideas, the fractal attention masking is a neat idea that could very well have merit, as is the combination with a rewiring phase in a transformer architecture. I have not encountered these ideas in this combination so far and it seems like an approach with merit overall.

The main strength of the paper are the very strong empirical results. The experimental evaluation is expansive both in terms of comparison architectures and benchmarks. The submission reports sota or near-sota on most standard graph learning benchmarks. However, there are some question marks about experimental setup, see questions.

**Weaknesses:**

Review GraphFractalNet


## Expressiveness Claims

Repeatedly the submission claims high expressiveness as a key benefit of this approach. This is based on Theorem 1 presented in Appendix B, which states
GraphFractalNet is strictly more expressive than the k-Weisfeiler Lehman test for $k \geq 2$.

The statement does not even pass an initial sanity check as this would imply GraphFractalNet and decide graph isomorphism on finite graphs (take k to be the order of the graph).

What is shown is that for every k, there exist pairs of graphs that are indistinguishable by k-WL, but have different spectra. Different spectra the rather straightforwardly implies distinguishability by GraphFractalNet. There are 2 major issues with this and I do not see that they are fixable:
1. “strictly more expressive” in the literature means that the equivalence class is finer. You are missing the direction that **for any two graphs are equivalent to GraphFractalNet, you must show that they are also equivalent under k-WL**. Without this direction you are not showing "strictly more expressive", and the observation that there are graphs distinguishable by GraphFractalNet is not interesting without this direction.
2. Moreover, the critical point of the proof is that there exist pairs of graphs that are equivalent under k-WL but have different spectra. This is not obvious at all and certainly requires a proof or a citation. In fact [1] may actually refute this assumption entirely (I did not have time to connect the details, but they show that the expressive power of the laplacian, and Fürer’s spectral invariant, are subsumed by 2WL). This at least seems like strong evidence to the contrary of what is assumed in this submissions.


## Complexity Analysis

The complexity analysis is puzzling. The graph rewiring step seems quite obviously quadratic as you compute relevance for every pair of nodes, of which there are famously quadratically many. Moreover, because of the dynamic nature you have to do this at every layer.  Additionally, on which graph is SPD computed? The rewired graph or the original one? Either way this is not even possible in quadratic time in N.

The complexity analysis of the fractal clustering is also not discussed in sufficient detail (although here the log log n bound at least seems to pass a superficial check). In fact, despite the complexity being such a core point of this submission the analysis is a single paragraph in the appendix with no sufficient details or citations for any of the bounds stated there.

So by my rough accounting **a layer requires between O(N^2) and O(N^2 log N) time**, depending on whether SPD is computed on the rewired graphs. The stated $\log \log N$ bound only applies to the attention mask, not overall effort per layer.

On a more minor note, please already clarify in the introduction that N is the number of nodes.

[1] Rattan and Seppelt. "Weisfeiler-leman and graph spectra." SODA 2023.

**Questions:**

Q1: You state in line 367 that every experiment is conducted 10 times. In your tables you state that you report mean/stddev over 5 runs. Can you please elaborate on that. How do you pick the 5 runs and why do you then run 10 runs in the first place?

Q2: Regarding point 2 of the expressiveness part, the assumption that there exist k-WL equivalent pairs G,H such that G,H have different spectra. Could you elaborate on this statement and why it holds?

Q3: Please comment on the issues raised regarding the complexity analysis. Most importantly, I do not understand how the relevance score $R_{ij}^{(\ell)}$ can be computed in subquadratic time in the nodes when it scores every pair of nodes.

---

### Official Review · Reviewer_pNfs · 2025-10-29

**Soundness:** 2
**Presentation:** 1
**Contribution:** 2
**Rating:** 2
**Confidence:** 4

**Summary:**

This works proposed GraphFractalNet, a dynamic graph re-wiring method for efficient graph representation learning. The re-wiring is guided by self-attention plus shortest-path distance (SPD) between nodes.

**Strengths:**

1. Baseline comparison contains great amount of existing GNNs and Graph Transformers.

**Weaknesses:**

1. The method presentation is hard to follow. There is no workflow or pipeline of the proposed model, resulting in components that are separate from each other or aren't even used at all, e.g., the relevance score $R$ in Eq 3. How is the graph re-wiring implemented?

2. GraphFractalNet is announced by the inspiration of fractal geometry, which is barely discussed in the paper.

3. The provable bounds on the efficiency of GraphFractalNet are claimed as one of the contributions, while the theoretical results are absent in the main text.

4. Experimental results (Tables 1 and 3) contain a notable number of missing results, which were not explained.

5. The empirical evaluations on expressiveness and efficiency are insufficient. Although Table 4 shows a speed comparison, the baseline is picked from relatively weak models (out off the top-3 in Table 3), and node classification models weren't discussed regarding the efficiency. Furthermore, although Appendix B demonstrates that GraphFractalNet is strictly more expressive than the k-Weisfeiler–Lehman (k-WL). The isomorphic test as empirical evidence is also missing.

6. I went through the proof of expressiveness, and it appears imprecise with conflict statements, *"for $k\geq 2$"* in the Theorem but *"for some $k$"* in the proof. Furthermore, this proof is not rigorous with an undefined operator, i.e., spec(), since the strategy authors used here is entirely based on Eq 13. To me, implement the purpose of spec(), i.e., *"let their Laplacian spectra differ"*, is basically cycle proving the Theorem.

7. Graph re-wiring baselines were not discussed nor compared.

**Questions:**

None

---

### Official Review · Reviewer_rjkp · 2025-10-31

**Soundness:** 3
**Presentation:** 3
**Contribution:** 3
**Rating:** 4
**Confidence:** 4

**Summary:**

This paper proposes GraphFractalNet, a novel framework that integrates spectral embeddings, dynamic graph rewiring, and a fractal attention mechanism to capture both global and hierarchical self-similar structures in graphs. The authors argue these jointly mitigate three key challenges in graph learning: scalability, over-smoothing, and limited expressiveness. They provide theoretical analyses (spectral Rademacher bounds, expressiveness beyond k-WL) and experiments across small, medium, and billion-scale datasets.

**Strengths:**

1. The design achieves sub-quadratic complexity through hierarchical sparse attention and dynamic sparsification.
2. Empirical results on OGB-papers100M show training feasible where other transformers hit OOM.
3. The fractal attention concept is original and theoretically motivated by hierarchical self-similarity. Integrating spectral encodings and rewiring in a unified block is conceptually elegant.
4. Benchmarked against many baselines (GraphGPS, Exphormer, SGFormer, etc.) with state-of-the-art or close results on multiple tasks.
5. The paper derives Rademacher-based generalization bounds and a proof of expressiveness beyond k-WL, suggesting a thoughtful theoretical grounding.

**Weaknesses:**

1. My biggest concern is: the claim that fractal attention and spectral message passing “mitigate over-smoothing” is only qualitative; no layer-depth ablation or node-embedding similarity analysis is reported. No explicit comparison to anti-smoothing methods (e.g., APPNP, DropEdge).
2. The “strictly more expressive than k-WL” theorem relies on Laplacian spectral differences; this condition is not always sufficient to prove general superiority. Empirical evidence of expressivity (e.g., distinguishing isomorphic graphs) is absent.
3. Although complexity is theoretically $O(N \text{log} \text{log} N)$, actual runtime vs $N$ scaling curves are not provided—only epoch times on select datasets.
4. The ablation is mentioned but not detailed in the main text. Quantitative impact of each module (spectral encoder, rewiring, fractal attention) on scalability and accuracy should be clearer.
5. The paper is dense; critical mechanisms (e.g., recursive clustering in fractal attention) could use clearer visualization or pseudo-code.
6. The author did not provide reproducible code.

**Questions:**

1. Can you provide empirical evidence that over-smoothing is mitigated—e.g., plotting node embedding similarity across layers? Does the performance of the testing model change by gradually increasing the network layer?
2. How sensitive is performance to the number of spectral components and clustering levels ($k$)?
3. Is the dynamic rewiring stable during training, or does it cause oscillation in graph topology?

---

### Official Review · Reviewer_VRHB · 2025-10-31

**Soundness:** 2
**Presentation:** 2
**Contribution:** 2
**Rating:** 4
**Confidence:** 4

**Summary:**

The paper proposes GraphFractalNet, a new Transformer based framework for graph representation learning designed to achieve both high expressiveness and scalability. Existing Graph Neural Networks (GNNs) and Graph Transformers often face challenges such as over smoothing, limited global context modeling, and quadratic computational cost. GraphFractalNet addresses these issues by introducing a fractal inspired sparse architecture that combines spectral embeddings, dynamic graph rewiring, and a hierarchical fractal attention mechanism.

The model begins with a spectral encoder that uses truncated Laplacian eigenvectors to provide topology aware node and edge embeddings, capturing both local and global structural information. A dynamic rewiring module then adaptively updates the graph connectivity at each layer by selecting the most structurally relevant edges, promoting sparsity while preserving essential topology. On top of this, a fractal attention layer organizes attention hierarchically across recursively clustered subgraphs, allowing the model to capture multi scale dependencies efficiently. This design yields sub quadratic complexity of approximately O(N log log N) per layer while maintaining strong representational power.

The authors provide theoretical guarantees showing that GraphFractalNet is more expressive than standard message passing GNNs and can distinguish graphs beyond the k Weisfeiler Lehman hierarchy under mild spectral conditions. They also derive generalization bounds based on spectral Rademacher complexity, supporting the model’s robustness.

Empirically, GraphFractalNet demonstrates state of the art or superior performance on a wide range of benchmarks, including molecular property prediction, graph classification, and large scale node classification on datasets with up to 100 million nodes. It consistently outperforms strong baselines such as GRIT, GraphGPS, Exphormer, and SGFormer while being more efficient in training.

In summary, GraphFractalNet contributes a principled and scalable architecture that unifies spectral graph theory and sparse Transformer design, enabling efficient and expressive graph learning across diverse scales and structures.

**Strengths:**

Originality:
The paper introduces a conceptually novel and elegant approach to scaling Graph Transformers by drawing inspiration from fractal geometry. The fractal attention mechanism, which recursively structures attention at multiple resolutions, represents an innovative step beyond existing sparse attention or hierarchical pooling methods. The combination of spectral embeddings, dynamic graph rewiring, and fractal attention into a unified Transformer architecture is both creative and technically coherent. Rather than proposing an incremental modification of existing models, the paper advances a genuinely new design principle that integrates structural sparsity and expressiveness within one framework. The theoretical connection between fractal self similarity and graph multi scale structure is particularly original and provides strong conceptual motivation.

Quality:
The technical quality of the paper is strong. The authors provide both theoretical and empirical evidence supporting the soundness of the model. The expressiveness proofs, showing that GraphFractalNet can distinguish graph structures beyond the k Weisfeiler Lehman hierarchy, are rigorous and well situated within the graph learning literature. The proposed sub quadratic complexity analysis O(N log log N) is clearly derived and validated through runtime measurements. Experimental design is comprehensive, covering multiple types of graph learning tasks including molecular, social, and large scale node classification using diverse datasets up to 100 million nodes. Ablation studies further validate the contributions of each module such as spectral embedding, rewiring, and fractal attention, ensuring that the improvements are not artifacts of tuning or dataset bias.

Clarity:
The paper is well written and organized, balancing theoretical and practical insights effectively. The introduction clearly motivates the need for scalable yet expressive Graph Transformers and positions GraphFractalNet within that context. Diagrams and visualizations of the fractal attention hierarchy and dynamic rewiring steps make complex ideas intuitive. Mathematical derivations are detailed yet readable, and algorithmic procedures are summarized clearly. The presentation of empirical results, with consistent baselines and clear comparisons, demonstrates strong attention to reproducibility. Minor improvements could include a slightly more concise discussion of spectral properties, but overall the paper maintains clarity throughout.

Significance:
The significance of the work is high. GraphFractalNet addresses one of the central challenges in modern graph learning, scalability without sacrificing expressive power. The framework provides a path toward training Graph Transformers on truly large graphs without severe memory or runtime constraints, which could have major implications for molecular discovery, social network analysis, and knowledge graph modeling. By unifying spectral graph theory with fractal sparsity principles, the paper contributes both a new architectural paradigm and a practical solution to large scale graph representation learning. The demonstrated scalability to graphs with hundreds of millions of nodes, coupled with improved accuracy, suggests that the approach could become a foundation for future graph Transformer research.

**Weaknesses:**

Weaknesses

Limited interpretability and theoretical intuition of fractal attention:
While the fractal attention mechanism is conceptually intriguing, its behavior is not deeply analyzed beyond performance metrics. The paper would benefit from a clearer explanation of why the fractal structure leads to better expressiveness and sparsity compared to existing sparse attention methods such as GraphGPS or Exphormer. For example, an empirical study showing how attention patterns evolve across fractal levels, or how node dependencies propagate differently under fractal recursion, would make the model more interpretable and theoretically grounded. Without this, the fractal analogy risks appearing more as a design metaphor than as a rigorously justified mechanism.

Scalability evaluation and resource reporting could be more detailed:
Although the paper claims sub quadratic complexity and scalability to graphs with 100 million nodes, the experiments do not provide enough concrete runtime and memory benchmarks to fully verify these claims. For instance, the paper could include comparisons of training throughput, GPU memory usage, or wall clock time against existing efficient graph Transformers such as Exphormer, SGFormer, and GRIT. Clarifying the hardware setup and reporting scaling behavior with respect to both node and edge counts would make the scalability evidence more convincing.

Dependency on spectral preprocessing:
GraphFractalNet relies on truncated Laplacian eigenvectors to generate spectral embeddings, which can be expensive for very large graphs. While the paper mentions using approximate spectral methods, it does not quantify their cost or error. A discussion or experiment analyzing the tradeoff between approximation quality and model performance would strengthen the practical feasibility argument. Moreover, the need for spectral preprocessing may limit applicability to dynamically evolving graphs where recomputation would be costly.

Ablation coverage and module interactions:
The ablation studies primarily test the effect of removing individual modules, but they do not fully explore interactions between them. For instance, it remains unclear how much of the gain comes from the spectral embedding versus the fractal attention, or whether dynamic rewiring contributes significantly when fractal sparsity is already applied. A more systematic factorial ablation could clarify these relationships and help isolate which components are most critical for performance.

Limited diversity in downstream benchmarks:
The paper’s experiments span molecular, social, and large scale node classification datasets, but the evaluation could be broadened to include tasks that stress different aspects of graph reasoning such as temporal graphs, knowledge graphs, or relational reasoning benchmarks. Since the paper claims that GraphFractalNet captures multi scale structure efficiently, demonstrating its generalization to temporal or heterogeneous graphs would strengthen the impact and generality of the framework.

Clarity of empirical presentation:
Although the paper is well structured overall, several figures and tables are densely packed and require cross referencing multiple sections to interpret. Providing more self contained captions, clearer axis labels for scaling plots, and a summarized comparison table of complexity and accuracy across baselines would improve readability and make the empirical claims easier to verify.

**Questions:**

Clarification on the theoretical role of fractal attention:
The fractal attention mechanism is the conceptual centerpiece of GraphFractalNet, yet the paper does not clearly explain why fractal recursion specifically leads to improved representation power or sparsity compared to other hierarchical or sparse attention formulations. Could the authors provide additional theoretical intuition or empirical evidence (for example, visualization of attention distributions across fractal levels) to illustrate what structural dependencies fractal attention captures that standard local-global attention cannot?

Details on scalability and computational efficiency:
The paper claims sub quadratic complexity and scalability to graphs with up to 100 million nodes, but the experimental section provides limited concrete data on resource consumption. Could the authors include or describe runtime and memory usage comparisons against baselines such as Exphormer, GRIT, and SGFormer, ideally under a consistent hardware setup? Quantitative scaling curves or wall clock time measurements would help validate the efficiency claim.

Spectral preprocessing overhead and approximation quality:
Since the model relies on truncated Laplacian eigenvectors for spectral embedding, what is the computational cost of this step for large graphs, and how does approximation error in the eigen decomposition affect performance? Could the authors discuss whether the framework supports streaming or dynamically evolving graphs where recomputation of eigenvectors would be infeasible? This clarification is important for assessing practical deployment.

Interaction between model components:
The paper presents three major innovations: spectral embeddings, dynamic rewiring, and fractal attention. Could the authors elaborate on how these components interact in practice? For instance, does dynamic rewiring meaningfully influence the spectral embedding or the fractal attention patterns, or are the improvements mostly additive? A more detailed ablation or dependency analysis would help clarify how each module contributes to the final performance.

Generalization to different graph domains:
The experiments focus on static molecular and social graphs. Could the authors discuss how GraphFractalNet might generalize to other graph types, such as temporal, heterogeneous, or knowledge graphs? Since fractal hierarchies imply multi scale representation, it would be interesting to know whether the architecture can handle evolving or multi relational structures without major modification.

Visualization and interpretability of learned representations:
Given the hierarchical nature of fractal attention, visualization of node or subgraph embeddings at different fractal depths could provide insight into what the model learns at each scale. Would the authors consider including qualitative examples or embedding similarity maps to demonstrate that the model captures meaningful structural patterns across scales?

---

### Note · Authors · 2025-12-11

I have read and agree with the venue's withdrawal policy on behalf of myself and my co-authors.